# Deciphering mixed infections by plant RNA virus and reconstructing complete genomes simultaneously present within-host

**Martine Bangratz[1]**, **Aurore Comte[1]**, **Estelle Billard[1]**, **Abdoul Kader Guigma[2]**, **Guillaume Gandolfi[1]**, **Abalo Itolou Kassankogno[2]**, **Drissa Sérémé[3]**, **Nils Poulicard[1]**, **Charlotte Tollenaere[1]** *

1 PHIM, Plant Health Institute of Montpellier, Univ. Montpellier, IRD, CIRAD, INRAE, Institute Agro, Montpellier, France, 2 INERA, Institut de l'Environnement et de Recherches Agricoles, Laboratoire de Phytopathologie, Bobo-Dioulasso, Burkina Faso, 3 INERA, Institut de l'Environnement et de Recherches Agricoles, Laboratoire de Virologie et de Biologie Végétale, Kamboinsé, Burkina Faso

☯ These authors contributed equally to this work.

* charlotte.tollenaere@ird.fr

**Data Availability Statement:** All obtained reads are available in European Nucleotide Archive (https://www.ebi.ac.uk/ena/browser/home), Bioproject:

## Abstract

Local co-circulation of multiple phylogenetic lineages is particularly likely for rapidly evolving pathogens in the current context of globalisation. When different phylogenetic lineages co-occur in the same fields, they may be simultaneously present in the same host plant (i.e. mixed infection), with potentially important consequences for disease outcome. This is the case in Burkina Faso for the rice yellow mottle virus (RYMV), which is endemic to Africa and a major constraint on rice production. We aimed to decipher the distinct RYMV isolates that simultaneously infect a single rice plant and to sequence their genomes. To this end, we tested different sequencing strategies, and we finally combined direct cDNA ONT (Oxford Nanopore Technology) sequencing with the bioinformatics tool RVhaplo. This method was validated by the successful reconstruction of two viral genomes that were less than a hundred nucleotides apart (out of a genome of 4450nt length, i.e. 2–3%), and present in artificial mixes at a ratio of up to a 99/1. We then used this method to subsequently analyze mixed infections from field samples, revealing up to three RYMV isolates within one single rice plant sample from Burkina Faso. In most cases, the complete genome sequences were obtained, which is particularly important for a better estimation of viral diversity and the detection of recombination events. The method described thus allows to identify various haplotypes of RYMV simultaneously infecting a single rice plant, obtaining their full-length sequences, as well as a rough estimate of relative frequencies within the sample. It is efficient, cost-effective, as well as portable, so that it could further be implemented where RYMV is endemic. Prospects include unravelling mixed infections with other RNA viruses that threaten crop production worldwide.

PRJEB76835, Biosamples: SAMEA115766186 to
SAMEA115766198, SRA accessions
ERS20300578 to ERS20300591.

**Funding:** This work was publicly funded by the
ANR (Agence Nationale de la Recherche, the
French National Research Agency) under the
EVCOPAR project (ANR-20-CE35-0004-01). The
funders had no role in study design, data collection
and analysis, decision to publish, or preparation of
the manuscript.

**Competing interests:** The authors have declared
that no competing interests exist.

## Introduction

The rise of high-throughput sequencing technologies is revolutionising our vision of global viral diversity, with an exponential increase in the rate of virus discovery in the past decade [1]. Embracing a virome approach (i.e. metagenomics analysis of viruses) has led to the characterization of viral diversity in natural ecosystems [2, 3], or the discovery of zoonotic agents with strong application for human health [4, 5]. For plant viruses, virome approach was applied to both cultivated and wild plants [6, 7], using different methodologies [8], such as the VANA (Virion-associated nucleic acid)-based _ metagenomics [9]. Recently, Oxford Nanopore Technology (ONT)-based methods are gaining more attention for viral genomics and metagenomics, due to the reduction of ONT sequencing errors, combined with their portability and low capital costs [10]. The use of ONT in plant virology is therefore slowly spreading throughout the plant virology community, with important applications for virus detection and surveillance, as well as whole genome sequencing [11–16]; as highlighted in the reviews on ONT and plant virus detection [17, 18]. These innovations have major implications for the risk assessment of new crop virus emergence and for the rapid development of effective disease control strategies.

In the case of strong local epidemics with active transmission and co-circulation of different phylogenetic lineages, the probability that a single plant is simultaneously infected by different viral haplotypes can be high (see e.g. [19]). Such mixed infections contrast with the case of any (single) viral infection, where within-host replication leads to the formation of viral clouds composed of closely related genetic variants, known as quasispecies [20]. Mixed infections involve more distantly related haplotypes, and in this case, within-plant dynamics of various haplotypes may differ from between-plant dynamics (transmission), with strong consequences for viral evolution [21–24]. Further exploration on these evolutionary and epidemiological consequences of mixed infections requires deciphering the genomes of the distinct viral isolates that simultaneously infect a plant [21, 25]. Classical molecular biology techniques (cloning and Sanger sequencing) may be used for this purpose, but they are time consuming, whereas ONT long reads provide an opportunity to access the multiplicity of genotypes within a plant sample and reconstruct within-plant diversity. However, this issue is not straightforward in terms of bioinformatic analysis, as it requires teasing apart highly similar sequences obtained from distinct genotypes from the same virus species. Recently, the growing interest in within-host viral diversity has led to the development of several bioinformatics methods to address this issue [26–29].

Rice yellow mottle virus (RYMV, genus Sobemovirus) causes an endemic rice disease in Africa that poses a threat to the sustainable development of rice production on the continent [30]. The virus is transmitted mechanically, with agricultural practices playing an important role, as well as through biotic means, including various insect vectors [31]. RYMV is a (+) single-strand RNA virus, with a genome of approximately 4450 nt, lacking a 3' polyA tail and organized into five overlapping open reading frames [30]. A recent study identified a rice yellow mottle disease hotspot in Burkina Faso [32, 33]. This irrigated site was found to harbor particularly high RYMV genetic diversity, with at least four distinct genetic groups coexisting over several years [33]. Such a context of disease and diversity hotspot would favor multiple infections (see above). Indeed, a fraction of samples (6/138, 4.3% of the dataset) analysed in this study presented mixed Sanger chromatogram (ORF4 gene coding coat protein), suggesting multiple independent infections, resulting in the simultaneous presence of various RYMV genomes within plant.

This study aims to identify a strategy for ONT sequencing library preparation and data analysis of the obtained reads to reconstruct the complete genomes of several distinct viral

haplotypes co-infecting a plant. First, we performed preliminary tests to assess the best methodology for preparing the libraries for ONT sequencing. Second, we generated artificial mixes to evaluate the performance of the proposed sequencing and bioinformatic approach. Finally, we applied the validated approach to field-collected samples to decipher mixed infections in the agrosystem. The methodology was also used to obtain full-length genomes of greenhouse isolates in a cost and time efficient manner.

## Methods

### Initial sequencing of single isolate to compare three different ONT sequencing strategies

Three different library preparation methods were tested. First, the ONT cDNA-PCR sequencing kit (SQK-PCS109) was used. For this, we first performed reverse transcription (RT), using the VN Primer from ONT Kit 5′-5phos/ACTTGCCTGTCGCTCTATCTTCTTTTTT TTTTTTTTTTTTTTTVN 3′, and simultaneously with two RYMV specific primers (position 1756 5′ACTTGCCTGTCGCTCTATCTTCCTCCCCCACCCATCCCGAGAATT 3′ and 4446 nucleotides 5′ ACTTGCCTGTCGCTCTATCTTCGGCCGGACTTACGACGTTCC 3′). After the denaturation step, strand-switching primers (SSP from ONT) 5′ TTTCTGTTGGTGC TGATATTGCTmGmGmG 3′ were added and incubated for 2 min at 42°C, followed by 1 h at 42°C after the addition of 1 μl of Maxima H Minus Reverse Transcription. The PCR was carried out using cDNA primers (cPRM from ONT) which attach the adapter primer link. We used Rapid Adapter (RAP) to the amplified cDNA library for sequencing with a Flongle Flow Cell R9.4.1 in a MinION MK1C. The sequencing duration ranged from 24 to 48 hours.

Secondly, a direct cDNA sequencing method was performed using the ONT Direct cDNA Sequencing Kit (SQK-DCS109) as described by [16]. Briefly, the extracted and DNase-treated RNA was ribodepleted using the QIAseq Fast Select rRNA Plant Kit (Qiagen Hilden, Germany). Double-stranded cDNAs were synthesised using random hexamers and the Maxima H Minus Kit (Thermo Scientific, K2561). The cDNA products were repaired and dA-tailed followed by ligation using the adapter MIX (AMX from ONT). A Flongle Flow Cell R9.4.1 was used for sequencing and the library was loaded as recommended by ONT (Flongle Sequencing Expansion Kit EXP-FSE002).

Each of the three methods (VN primers, RYMV-specific primers and direct cDNA) was performed twice, using RNA extracted (GeneJET Plant RNA Purification Mini Kit, Thermo Scientific, K0802) from rice plant infected with the BF710 isolate. This infected material came from experimental infections carried out mechanically under controlled conditions [34], using symptomatic rice leaves collected two to five weeks post-inoculation in greenhouses. This approach for viral replication was employed for all samples referred to as "isolates." The initial inoculum consisted of rice leaves collected from fields in southwestern Burkina Faso (see Supporting Information S1 Table, and [33]). During field surveys, farmers were individually asked for permission to sample leaves from their fields. The entire project adhered to the Nagoya protocol guidelines on access and benefit-sharing.

Basecalling was performed using guppy 6.3.7 [35]. To visualise the coverage of RYMV for each run, we use the mapping tool Minimap2 v2.24 [36] and the samtools suite v1.10 [37] with the isolate used for the run as a reference.

### Sequencing of artificial mixed-infection samples

Total RNA was extracted from rice leaves infected by different RYMV genotypes, using the RNeasy Plant Mini Kit (Qiagen, Hilden, Germany). For the preparation of artificial mixes, we

selected a set of three RYMV genotypes from Burkina Faso (S1 Table): BF710, BF711 (Banzon, collected in 2016) [32], and BF706 (Karankasso Sambla, collected in 2014) [38]. The full-length genomes of these three isolates were obtained by Sanger sequencing. Their divergence ranges from 2% (110 nucleotides) (BF710 and BF711) to 3% (131 nucleotides) (BF710 and BF706).

A set of five artificial mixed samples was performed, containing two of the three genotypes in different relative proportions (Table 1). For this purpose, estimation of RYMV viral load in samples is a mandatory step, and was performed by qRT-PCR according to the protocol previously described with slight modifications [39]. From 1 µg of total RNA extract from leaves infected respectively by BF706, BF710, BF711 genotypes, three reverse transcriptions were performed with the reverse primer R0 (nucleotides 1748–1767: 5'-GGCCGGACTTACGAC GTTCC-3'). 5 µl of each cDNA (diluted 1:1250) were followed by three qPCR (nine qPCR reactions in total per sample) mixed with 12.5 µl of Full Velocity Master Mix (Stratagene) and 300 nM sense and antisense primers (FqPCR2 sense primer, nucleotides 673–690 5' -ACCT CCTCATCGTCTTGG- 3' and RqPCR2 antisense primer nucleotides 1049–1064 5' -CGGC GGCACATCTTCG- 3'). The number of copies per 1 µg RNA was calculated. Dilutions were performed to adjust the concentrations to 4,3 x $10^{-7}$ copies. Artificial mixes with the desired relative proportions were then obtained: 50/50 in some cases and 99/1 in others. They were sequenced using the "direct cDNA" method, as were the two original isolates BF711 and BF706.

## Haplotype reconstruction from artificial mixes

Basecalling was performed with guppy 6.3.7 [35]. Then, after preliminary evaluation of several methods, AssociVar [27], Nano-Q [29], Variabel [40], we chose the bioinformatics tool RVHaplo [26] for viral haplotypes reconstruction from ONT sequencing of mixed infections.

A snakemake pipeline was created to launch several datasets simultaneously (https://forge. ird.fr/phim/acomte/rvhaplosmk) and to favour reproducibility. For each dataset, after initial testing of various thresholds, we decided to remove from the datasets (prior to analysis) all reads shorter than the N50 (i.e. the length of the read found at the midpoint of the length-order concatenation of all obtained reads; or in other words: the length of the shortest read in the group of longest sequences that together represent half of the nucleotides in the set of sequences). This led to speed up the calculations, and obtained data were more consistent with expectations.

## Sequencing field-collected samples

Four rice leaves samples, symptomatic for RYMV, were collected in the irrigated perimeter of Banzon (south-west Burkina Faso) and selected for the application of the proposed methodology because the amplification of the RYMV ORF4 (coat protein gene) Sanger sequencing presented a mixed chromatogram. First, the two samples MP1458 and MP2967, collected in 2017–2018 in Banzon (see doi.org/10.23708/8FDWIE [32] and S1 Table), and already studied in terms of OFR4 sequencing in [32] were selected. These samples were dried right after sampling, and then stored at ambient temperature [33].

We then analysed two rice leaf samples, EF0750 and EF0846, collected in 2021 (S1 Table), which also showed mixed chromatograms when sequencing ORF4 by the Sanger method (unpublished data). Sample preservation was better for these two samples collected in 2021 (they were kept dried and then frozen at -20 a few weeks after sampling), compared to older (2017–2018) samples (kept dried at ambient temperature for several months or even years).

**Table 1. List of the 13 nanopore runs performed with the selected ONT sequencing strategy, with the results obtained, in terms of basic sequencing information (after basecalling) as well as haplotype reconstruction (using RVhaplo analysis).**

| Run | Initial Sample information | | | | Type of Flongle | Sequencing information | | | | | RVhaplo results | | | |
|---|---|---|---|---|---|---|---|---|---|---|---|---|---|---|
| | Sample code | BF710 | BF706 | BF711 | | number of reads | number of nucleotides | Mean read quality | average read lenght | N50 | haplotypes | Sequence lenght (nt) | proportion | distance to reference (nt) |
| 1 | **MixA** | 1% | 99% | - | R9.4.1 | 329 297 | 150 218 393 | 10.0 | 456.2 | 543 | h0 | 4426 | 0.99 | BF706 (2) |
| | | | | | | | | | | | h1 | 4408 | 0.01 | BF707 (0, ORF4 only)/ BF710 (192) |
| 2 | **MixB** | 99% | 1% | - | | 919 987 | 513 698 291 | 10.0 | 558.4 | 688 | h0 | 4442 | 0.98 | BF710 (1) |
| | | | | | | | | | | | h1 | 4416 | 0.02 | BF706 (2) |
| 3 | **MixC** | 50% | 50% | - | | 45 752 | 12 864 675 | 8.5 | 281.2 | 352 | h0 | 4400 | 0.71 | BF706 (3) |
| | | | | | | | | | | | h1 | 4398 | 0.29 | BF710 (3) |
| 4 | **MixD** | 1% | - | 99% | | 413 757 | 120 306 275 | 10.1 | 290.8 | 324 | h0 | 4427 | 0.99 | BF711 (0) |
| | | | | | | | | | | | h1 | 1983 | 0.01 | BF710 (7) |
| 5 | **MixE** | 99% | - | 1% | | 795 581 | 315 914 275 | 11.1 | 397.1 | 488 | h0 | 4443 | 0.98 | BF710 (1) |
| | | | | | | | | | | | h1 | 4410 | 0.02 | BF711 (0) |
| 6 | **BF706** | - | 100% | - | | 395 390 | 156 317 339 | 11.3 | 395.3 | 492 | h0 | 4441 | 0.99 | BF706 (2) |
| | | | | | | | | | | | h1 | 4404 | 0.01 | BF707 (0, ORF4 only) / BF706 (161) |
| 7 | **BF711** | - | - | 100% | | 583 722 | 194 193 438 | 10.7 | 332.7 | 408 | h0 | 4443 | 1.00 | BF711 (0) |
| 8 | **MP1458** | Unknown (Field samples) | | | | 37 176 | 8 838 182 | 10.0 | 237.7 | 270 | h0 | 4418 | 1.00 | NA |
| 9 | **MP2967** | | | | | 9 456 | 2 509 759 | 9.2 | 265.4 | 293 | h0 | 4380 | 0.80 | |
| | | | | | | | | | | | h1 | 4041 | 0.20 | |
| 10 | **EF0750** | | | | | 967 250 | 491 740 884 | 10.0 | 508.4 | 659 | h0 | 4446 | 0.67 | |
| | | | | | | | | | | | h1 | 4440 | 0.33 | |
| 11 | **EF0846** | | | | | 1 001 106 | 505 134 981 | 10.6 | 504.6 | 668 | h0 | 4446 | 0.40 | |
| | | | | | | | | | | | h1 | 4444 | 0.36 | |
| | | | | | | | | | | | h2 | 4443 | 0.24 | |
| 12 | **GH batch 1** | Mix of 3 or 4 partially known isolates | | | R10.4.1 | 481 881 | 282 483 053 | 9.8 | 586.2 | 719 | h0 | 4427 | 0.42 | EF0580 (0, ORF4 only) |
| | | | | | | | | | | | h1 | 4444 | 0.35 | EF0768 (0, ORF4 only) |
| | | | | | | | | | | | h2 | 4426 | 0.22 | EF0252 (4, ORF4 only) |
| 13 | **GH batch 2** | | | | | 1 047 232 | 588 799 993 | 10.5 | 562.2 | 684 | h0 | 4443 | 0.32 | EF0316 (0, ORF4 only) |
| | | | | | | | | | | | h1 | 4426 | 0.30 | EF0321 (0, ORF4 only) |
| | | | | | | | | | | | h2 | 4426 | 0.23 | EF0644 (0, ORF4 only) |
| | | | | | | | | | | | h3 | 4441 | 0.15 | EF0562 (0, ORF4 only) |

For each run of direct cDNA sequencing from RT with random hexamer primers after ribodepletion (13 in total), the initial sample information, the type of flongle (R9.4.1 or R10.4.1), basic sequencing quality information and finally, the results obtained from the RVhaplo analysis are given.

The "initial sample information" section indicates whether the sample corresponds to 1) one known RYMV isolates singly sequenced in isolation (runs 6 and 7) or as a mixture of two isolates with known proportions (runs 1 to 5).

The sequencing information section includes the number of reads obtained, the total number of nucleotides obtained, the average read quality (QC), the average read length and N50. N50 is the length of the shortest read in the group of longest sequences which together represent half of the nucleotides in the sequence set. All reads obtained are available in the European Nucleotide Archive (https://www.ebi.ac.uk/ena/browser/home), Bioproject: PRJEB76835, Biosamples: SAMEA115766186 to SAMEA115766198, SRA accessions ERS20300578 to ERS20300591.

The results of the RVhaplo analysis include the list of haplotypes obtained (from 1 to up to 4 haplotypes) with their sequence length, the proportion of each haplotype in the sample, and the closest known genome isolate (together with the number of nucleotides that differ between the haplotype obtained and the known isolate).

## Sequencing greenhouse samples

We then aimed to obtain the full-length genomes of seven RYMV isolates originally collected in Burkina Faso in 2021, and used in greenhouse experiments, for which a partial sequence (ORF4, Coat Protein gene) was available by Sanger sequencing. For this purpose, and to be cost and time efficient, we arranged the seven isolates into two batches: EF0252, EF0580, and EF0768 in "Greenhouse batch 1", and EF0316, EF0321, EF0562, EF0644 in "Greenhouse batch 2". These isolates all originated from a sampling performed in 2021 in two sites from western Burkina Faso: Banzon and Badala (S1 Table). They were amplified on rice (cultivar IR64) grown in greenhouses.

The direct cDNA sequencing kit (SQK-DCS109) is no longer available since June 2023. As recommended by ONT, we then used their latest Q20+ chemistry and the V14 ligation sequencing kit (SQK-LSK114). We produced double-stranded cDNAs followed by adapter ligation using the ONT protocol ligation sequencing kit V14 and used a Flongle Flow Cell R10.4.1 for the sequencing step. We then analyzed obtained data as described above.

## Phylogenetic analysis of the genomes obtained

The different nanopore runs performed resulted in a total of 28 sequences, all but one being longer than 4 000nt (see Table 1, and S1D Fig). All 27 near full-length sequences were aligned to the three reference genomes obtained by Sanger sequencing. A Neighbor-Joining (NJ) phylogenetic tree was then constructed using MEGAX [41] to represent the diversity obtained from the analysed samples.

These sequences were then assigned to 19 distinct genome sequences. These newly obtained sequences were then compared with 47 full-length genome sequences from isolates collected in West and Central Africa, available in NCBI (July 2023). Maximum-Likelihood (ML) phylogenetic tree was reconstructed using the best-fitting substitution model (GTR+G+I) determined with MEGAX, with 100 bootstrap replicates. Phylogenetic trees were drawn using FigTree v1.3.1 (http://tree.bio.ed.ac.uk/software/figtree/). An estimate of genetic diversity was obtained from MEGA for the newly obtained sequences, compared to the sequences available in the literature.

Finally, potential recombination signals from the complete RYMV genome sequence dataset (*i.e.* 66 RYMV complete genomes, 19 obtained during this study and the 47 available in public database) were searched using the seven algorithms implemented in the RDP program v4.101 [42]. Only recombination events detected by at least four methods and with *P*-values below $10^{-2}$ were considered.

# Results

## Comparison of three ONT sequencing strategies

Fig 1 shows the results obtained for the three sequencing strategies, in terms of genome coverage, number of reads, read length, proportion of reads mapped to the RYMV genome, for the six runs performed on the same RYMV isolate (BF710). The direct cDNA sequencing approach, following [16], yields the highest percentage of reads mapped to the RYMV genome (95% compared to 19–47%, on average 33% for the other two methods). The average read length was also much higher for this method (on average 648 nucleotides) compared to the others (335–423 nucleotides). Although the total number of reads obtained was lower than that using the VN primers (oligo-dT) technique, direct cDNA sequencing gave by far the best results in terms of genome coverage (Fig 1) and was selected for subsequent runs.

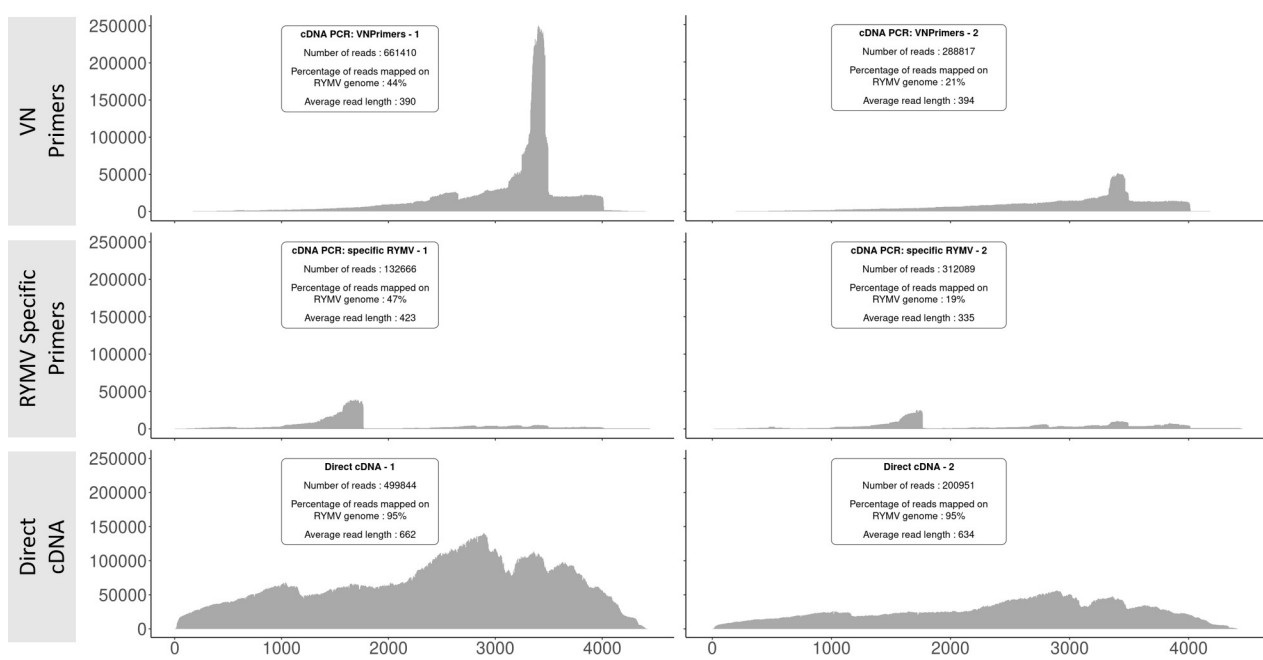

**Fig 1. RYMV genome coverage obtained for three ONT sequencing strategies performed on the same sample.** Comparison between the three methodologies for RYMV sequencing of the BF54 isolate, with two Oxford Nanopore Technology (ONT) sequencing replicates (left and right panels of the figure) for each of the three methods: amplification with VN Primers on top, RYMV specific primers on the middle, and direct cDNA sequencing on the bottom.

It should be noted that RYMV often encapsulates a small circular satellite RNA [30], which has been reported in all isolates originated from West and Central African [43]. This 220nt-long sequence was not found in our ONT sequences, although it could be detected by specific PCR in some isolates analysed.

## Analysing the results of artificial mixes results to assess the bioinformatic pipeline

We then tested whether we could recover the different RYMV haplotypes in artificial mixes of two RYMV isolates. For this purpose, we used two additional RYMV isolates (BF706 and BF711), also originating from Burkina Faso, in addition to the reference isolate BF710 mentioned above. Table 1 shows the results obtained for the five artificial mixes and the two isolates BF706 and BF711, all performed with the direct cDNA sequencing approach.

We sequenced samples (mixA-mixE) obtained from these isolates, previously quantified, and mixed in different relative proportions of two isolates: equal proportion (50–50%, mixC), or a major isolate in mixed infection (99–1% mix A, B, D, E) (these artificial mixes correspond to runs 1–5 in Table 1). We then sequenced the two isolates BF711 and BF706 taken separately (see runs 6–7 in Table 1). For these 7 runs, the number of reads ranged from 45 752 to 919 987 (mean = 497 640, Table 1 and S1B Fig), and the average read length ranged from 281 to 558 (mean = 387) (Table 1).

For the artificial mixes (runs 1 to 5), RVHaplo was able to identify the multiplicity of infection and to reconstruct the genomes in most cases. For the 50/50 (mixC, run 3), we obtained the two expected genomes (approximately 4400nt) each with only 3 differences from Sanger sequences, and RVhaplo estimated the relative proportions to 70/30 (Table 1). For the other mixes, all with relative proportions of 99/1, we obtained the expected haplotypes in most cases,

with full length genome reconstruction and relative estimate 99/1 or 98/2. This is the case for mixB (run 2) and mixE (run 5). For mixD (run 4), it was only partially true as the minority haplotype was neither complete, nor perfectly matched (BF710: 1983nt, 7 errors); a result most likely attributable to the lower quality of this specific run (N50 = 324, Table 1 and S1C Fig).

On the other hand, for mixA, we obtained the sequence of the majority isolate (BF706), but the minority (1%) haplotype does not match the minority haplotype included in the mix (BF710). Instead, its ORF4 partial sequence corresponds exactly to another isolate manipulated in greenhouses at the same time (BF707; [36]). When the isolates were sequenced independently (runs 6–7), we obtained exactly the expected haplotype (4443nt) for isolate BF711 (run 7), but two distinct isolates for BF706 (run 6). In fact, the first haplotype corresponds to the expected isolate BF706 (with only 2nt difference out of a total of 4441nt), while the second unexpectedly corresponds to a haplotype distant from 161nt from BF706 (run 6), but matching the isolate BF707 over the ORF4 partial sequence. We therefore interpret the results obtained for both run 1 and 6 as contamination of BF706 by BF707 during leaf manipulation (as these two isolates were not manipulated together in the laboratory, but in the greenhouse instead). For MixA (run 1), the minor isolate (BF710) could be recovered by changing the read length threshold applied prior to data analysis (set to N50 for all analyses presented here). Indeed, when the threshold was set to 700 instead of 543, three haplotypes were deciphered: BF706 (98%), BF707 (1%) and BF710 (1%), all three with a minimum length of 4336, and a maximum difference of 2nt from the Sanger reference.

Overall, we appreciate the efficiency of the approach in deciphering the haplotypes present in artificial mixes, so these results validate the approach for library preparation and bioinformatic analysis. The phylogenetic tree (S1E Fig) clearly shows a difference between the different haplotypes (distance from 110 to 192nt, 2–4% of the genome), compared to the artefactual haplotypes (maximum 7nt difference). These runs of known composition therefore allow to estimate a range of thresholds for the divergence of haplotypes that could be deciphered by the approach: mixed infections with genomes differing by more than 2% should be retrieved by the proposed methodology, while it would most likely not work for genomes differing by less than 0.05% of their genome.

## Deciphered mixed infections and newly obtained full-length genomes

Once the sequencing strategy and the bioinformatics tools were validated, we applied the approach to a set of four field samples, namely rice leaves collected in Burkina Faso. While the two samples collected in 2017–2018 (MP1458 and MP2967, runs 8–9 Table 1) yielded lower quality results (average read length maximum 265), most likely due to a conservation issue, each of the two samples collected in 2021 (EF0750 and EF0846, runs 10–11) resulted in good overall sequencing (about $10^6$ reads, N50 about 650nt, see Table 1, S1B and S1C Fig). While one sample (MP1458) yielded only one haplotype, the other three resulted in various haplotypes (ranging from 110 to 265nt divergence within each sample), with up to three complete (at least 4443nt) genomes recovered in the 40/36/24 ratio from sample EF0846 (run 11, Table 1). Combining the results of these four runs performed on field samples, we obtained a total of eight new RYMV genomes that could be placed in a phylogenetic tree (Fig 2, six haplotypes in green). It should be noted that one of these haplotypes (2018MP2967_h1) was truncated in its 5' part due to a lower quality area (10 gaps in ca 150nt area) detected when aligning to the sequences from the literature. This haplotype is the minority one from a lower quality sample and all other sequences were fine.

In addition, we used the same strategy to obtain seven full-length genomes from two flongle runs (runs 12 and 13, Table 1) performed on artificial batches comprising three to four isolates.

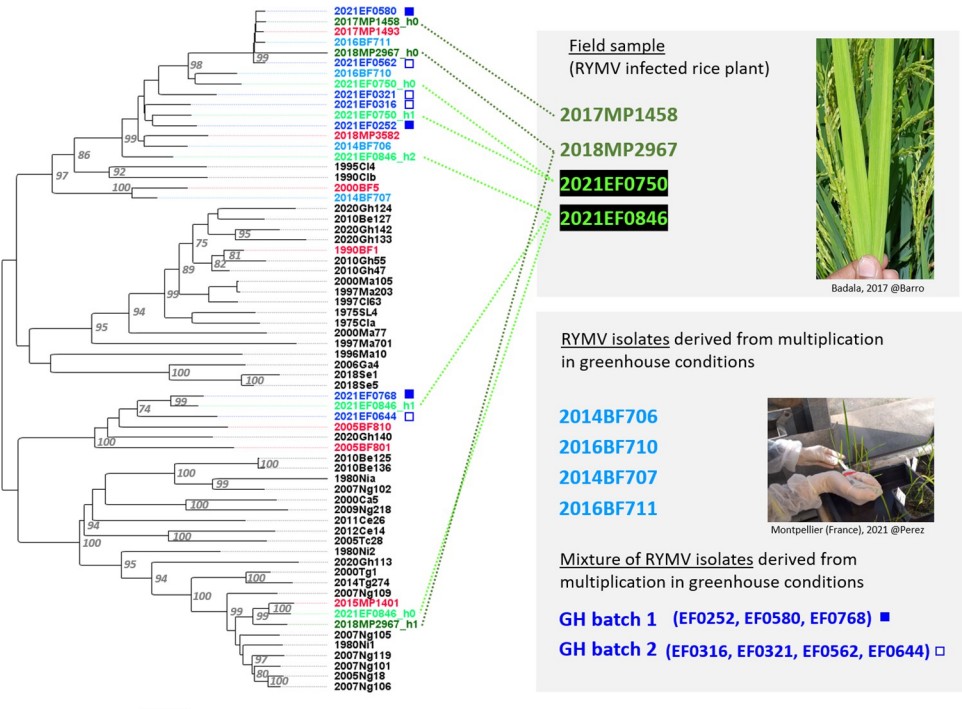

**Fig 2. Phylogenetic tree replacing the 19 RYMV haplotypes obtained by nanopore sequencing (this study) within the 47 previously published viral genomes from samples collected in West Africa (literature).** Maximum likehood (ML) phylogenetic tree of 47 RYMV genomes of isolates from West and West Central Africa (NCBI database) together with the 19 haplotypes from this study. The ML phylogenetic tree was constructed from the full-length genome sequences (4461nt). Greenhouse isolates are shown in blue (light blue for the isolates sequenced individually, and dark blue for the batches of 3 or 4 isolates sequenced altogether), while the field isolates are represented in green (dark green for samples collected in 2017–2018, and light green for others, collected in 2021). The images illustrate the difference between these two types of samples (field samples and viral isolates derived from amplification under controlled conditions). Literature sequences also originating from Burkina Faso are shown in red, while sequences derived from samples collected in other countries are shown in black.

The sequences obtained were analysed using the developed pipeline, resulting in the expected number of distinct haplotypes for both runs. By comparing the obtained genomes with partial ORF4 sequences (Sanger sequencing), we were able to retrieve the full-length genomes of the seven different isolates (Fig 2, haplotypes in blue, marked with stars).

The phylogenetic analysis of 66 genomes (Fig 2) shows the contribution of these nanopore sequences to the genomic diversity of RYMV in West Africa: 19/66 = 28.8% in terms of number of viral genomes. Although the sampling is local (south-western Burkina Faso), this area being a hotspot of RYMV diversity [32], this dataset represents a substantial contribution to the genetic diversity of RYMV in West Africa. Indeed, the genetic diversity of these 19 new sequences is 0.043 ± 0.002 substitutions/site, compared to 0.065 ± 0.003 subst./site for the 47 sequences from the literature (overall genetic diversity: 0.062 ± 0.003 subst./site). Finally, the recombination analysis (RDP4, see S2 Table) detected a new recombination event in Burkina Faso, in addition to the S1bzn group reported in [32], so that at least two recombinant haplotypes circulate within the irrigated perimeter of Banzon.

## Discussion

Local co-circulation of multiple phylogenetic lineages, and within-plant viral diversity is particularly likely for rapidly evolving pathogens such as RNA viruses [44]. Mixed infections, and

subsequent virus-virus interactions, are likely to affect viral diversity and evolution [45]. In such a context, long reads generated by ONT seem promising for deciphering the multiple viral genomes simultaneously infecting the same host. Indeed, our results show that it is possible to use direct cDNA ONT sequencing in combination with the bioinformatics tool RVhaplo, to identify different haplotypes of the rice yellow mottle virus (RYMV) co-infecting a single rice sample, and to obtain their full-length sequence. This was true even for 99/1 mixes of isolates, that differed in only 2–3% of their genomes (about a hundred nucleotides out of viral genome of about 4450nt).

The method described by Liefting et al [16], based on direct cDNA sequencing, efficiently yielded almost full-length RYMV genomes, with the sequences obtained being either exact, or very close (i.e. 2nt out of 4450nt genome) to the same samples sequenced by Sanger method. The proportion of reads mapping to the RYMV genome was much higher with this method (95%) than with PCR-based strategies, both using VN primers, or RYMV-specific primer (range from 19 to 47% of reads mapped on viral genome). Average read lengths were also higher with the cDNA direct sequencing strategy, which was consequently selected to go further in sequencing mixed-infected samples.

However, we note that the sequencing method based on reverse transcription with polyT primers works even though RYMV does not have a 3' polyA tail. This has also been observed for other viruses (e.g. Potato leafroll virus [14]). Here, this result can be explained as the repartition of reads along the RYMV genome is highly aggregated on A-rich regions of the genome. Although this strategy was left up for the purpose of this study, the information on the possibility of sequencing RYMV from polyT primers RT may be useful, e.g. for metagenomics studies.

The bioinformatic tool RVhaplo [26] was used to analyse the data obtained through cDNA direct sequencing (see above), and allowed an efficient reconstruction of the different haplotypes present as mixed infections in our study. This was true even in the case of a mixed infection where the majority isolate was 99%, mixed with only 1% of another isolate, demonstrating the sensibility of this approach. Also illustrating the high sensibility of the method, is the detection of an unexpected haplotype in one of the isolates multiplied under controlled conditions (greenhouses). Representing only 1% of the sample analysed, this haplotype would probably have remained undetected by Sanger sequencing, so that an application of the methodology presented here is also to screen a sample for potential contamination prior to setting up sensible greenhouse experiments.

The methodology described is rapid and appears to be cost-effective. Indeed, in this work, the use of two Flongles resulted in seven genomes for greenhouse isolates, and we also generated five genomes from two Flongles used on two field samples. Including the reagents (in particular the ribodepletion step), price per viral genome of the proposed approach is comparable to Sanger in our conditions. Globally, this study generated 19 full-length genomes, compared to a total of 47 available to date for RYMV in West Africa, so it adds to the knowledge in terms of full-length genomes in this area. This is particularly relevant in the context of viral diversity hotspots, such as in western Burkina Faso, where distant genetic groups co-circulate, providing an opportunity for recombination [33]. Other recombinant isolates have previously been reported for RYMV in East Africa [46], the center of origin of RYMV, where RYMV genetic diversity is highest. Also, viral epidemics in Madagascar have been shown to result from a single introduction of a lineage potentially resulting from a recombination event [47]. Recently, increased RYMV dispersal has led to a reduction in the historically strong geographic structuration of viral diversity in West Africa [33, 48, 49]. Such a context is likely to increase the risk of recombination between distant genetic lineages. In western Burkina Faso, a recombinant lineage has already been evidenced in a previous study [33], and the additional genomes obtained in this study allowed the identification of a second recombination event. Further

ONT sequencing of RYMV genomes would likely lead to the detection of others, especially as the period of co-circulation of lineages becomes longer. This potentially increases risks in terms of resistance breakdown and pathogenicity [50].

Deciphering the different genomes simultaneously present in a host in the context of mixed infection remains a methodological challenge. In our study, we were able to decipher mixed infections of rice infected by various RYMV isolates, even for mixes in 99/1 ratio, and two isolates that differ in only from 2–3% of their genomes (about one hundred nucleotides out of a viral genome of about 4450nt). This was most likely made possible by the use of a long-read sequencing approach, as reported in the detection of two strains of DNA virus [11], or using PacBio long-read HiFi sequencing to unravel intra-host diversity across natural RNA mycovirus infections [51]. Previous attempts to characterize within plant diversity in RYMV using a short-read strategy (Illumina technology) allowed to identify that the sample corresponded to a mixed infection, and to estimate the proportion for each polymorphic site, but it could not phase the polymorphic sites and reconstruct the distinct viral genomes (Poulicard et al, unpublished). On the other hand, sequencing of virus-derived small interfering RNA (vsiRNA) led to the distinction of several strains of three different viral species in strawberry [52]. A recent systematic comparison of ONT and Illumina sequencing from total RNA (unfortunately not including the cDNA-direct sequencing method used in this study) showed the performance of ONT sequencing in terms of quality, affordability and practicality, for different viral organization and including low titer virus [53]. In this study, the high titer of RYMV in rice (strong within plant accumulation [39]) may have helped, and the applicability of the described method to other RNA viruses remains to be evaluated.

Such mixed viral infections are important because interactions between viral genomes ('social life of viruses', [24]) can influence the ecology and evolution of plant diseases [21, 25]. For fungi, Lopez-Villavicencio et al [54] showed that the fungal genotypes found in each plant are more related than expected by chance, suggesting that *Microbotryum violaceum* actively excludes dissimilar genotypes while tolerating closely related competitors. The method described in this article allows to distinguish genomes with a distance of 2% (110nt out of about 4450nt). On the other hand, the viral clouds of highly related genetic variants generated by within-host replication in high mutation rate RNA genomes are not captured by the proposed methodology. Indeed, although nanopore single molecule sequencing technology is able to generate reads of the different variants from the viral cloud, such sequences, each of which is extremely rare and differs at very few positions, were not retrieved after bioinformatic analysis. This is in line with the aim of this study, which is to decipher the supposedly independent infections by different RYMV isolates.

ONT sequencing is a highly dynamic methodology (as exemplified here by the switch from flow cells R9 to R10 during the course of this study _ R9 was no longer available). Its flexibility and diversity of library preparation make it adaptable to different context (as a 'swiss knife'). This combined with its practicality and performance makes ONT sequencing an ideal tool for rapid and affordable virus detection and genomics [53]. It will continue to evolve, and undoubtedly improve in the near future, making it a promising technology not only for viral genomics but in general [55]. As a portable sequencing technology [12], requiring no costly initial investment and being cost-effective compared to Sanger or Illumina, it holds great promise for the global south, and particularly Africa. Further deployment of ONT sequencing in Africa requires capacity building, mainly in terms of bioinformatic resources and skills, and various initiatives are underway to achieve this goal (in Mali [56], or Burkina Faso, pers.com.). Such capacity building is required to better understand and fight crop diseases around the world, a matter of paramount importance given that the greatest losses frequently corresponds to emerging diseases in food-deficit regions with rapidly growing populations [57].

## Supporting information

**S1 Table. List of RYMV-infected rice leaves samples from Burkina Faso analyzed in this study.** The site of initial sample collection IS indicated along with the ID of the field, GPS coordinates and date of sampling. Genbank accession numbers are also mentioned for all the sequences obtained.
(DOCX)

**S2 Table. Detection of recombination events.** Two recombination events were detected using the RDP4.101 software. The first one was found in only one RYMV haplotype, and the other, already documented in Billard et al 2023 [32], was found in eight viral haplotypes. For each recombination event, we indicate the most likely parents (major and minor), the breakpoints (with their range of confidence in brackets), and the methods detecting the recombination events with the associated *P*-value.
(DOCX)

**S1 Fig. Results of the 13 ONT runs performed using the cDNA direct sequencing strategy (see also Table 1).** A. Total number of bases obtained for each run (log scale). B. Total number of reads obtained for each run (log scale). C. N50 for each run (defined as the length of the shortest read in the group of longest sequences that together represent half of the nucleotides in the set of sequences). D. Sequence length of the 28 different haplotypes reconstructed by the RVhaplo method over the 13 runs. E. Neighbor-joining phylogenetic tree of the 27 (almost full length) haplotypes obtained, with the four reference genomes obtained by the Sanger technique. The legend, common to the different panels, appears on the bottom right and distinguishes: in green, the field samples, and in blue, the samples corresponding to viral amplification in greenhouses (and then either sequenced either in isolation, or in mixes).
(TIF)

## Acknowledgments

This work was carried out within the formalized partnership "International joint Laboratory LMI PathoBios: Observatory of plant pathogens in West Africa: biodiversity and biosafety" (www.pathobios.com; twitter.com/PathoBios). We thank the rice farmers from the study sites in Burkina Faso for their kind collaboration. We also thank Dr Cai for helpful advice at early stages of bioinformatic analyses. We thank Agnès Pinel-Galzi for sharing protocols, especially for satellite RNA detection. The manuscript benefited from helpful comments by Eugénie Hébrard. The authors acknowledge the ISO 9001 certified IRD itrop HPC (member of the South Green Platform) at IRD Montpellier for providing HPC resources that have contributed to the research results reported within this paper, URL: https://bioinfo.ird.fr/, http://www.southgreen.fr.

## Author Contributions

**Conceptualization:** Martine Bangratz, Aurore Comte, Estelle Billard, Nils Poulicard, Charlotte Tollenaere.

**Data curation:** Martine Bangratz, Aurore Comte, Guillaume Gandolfi.

**Formal analysis:** Martine Bangratz, Aurore Comte, Estelle Billard, Guillaume Gandolfi, Nils Poulicard.

**Funding acquisition:** Nils Poulicard.

**Investigation:** Martine Bangratz, Aurore Comte, Estelle Billard, Abdoul Kader Guigma, Guillaume Gandolfi, Abalo Itolou Kassankogno, Drissa Sérémé, Charlotte Tollenaere.

**Methodology:** Martine Bangratz, Aurore Comte, Estelle Billard, Guillaume Gandolfi.

**Project administration:** Abalo Itolou Kassankogno, Drissa Sérémé, Charlotte Tollenaere.

**Resources:** Charlotte Tollenaere.

**Software:** Aurore Comte, Guillaume Gandolfi.

**Supervision:** Martine Bangratz, Aurore Comte, Charlotte Tollenaere.

**Validation:** Abdoul Kader Guigma, Nils Poulicard, Charlotte Tollenaere.

**Visualization:** Estelle Billard, Nils Poulicard.

**Writing – original draft:** Martine Bangratz, Aurore Comte, Nils Poulicard, Charlotte Tollenaere.

**Writing – review & editing:** Martine Bangratz, Aurore Comte, Estelle Billard, Abalo Itolou Kassankogno, Nils Poulicard, Charlotte Tollenaere.

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
