## [Decision Letter · Decision Letter 0]

22 May 2024

PONE-D-24-16411Deciphering mixed infections by plant RNA virus and reconstructing complete genomes simultaneously present within-hostPLOS ONE

Dear Dr. Tollenaere,

Thank you for submitting your manuscript to PLOS ONE. After careful consideration, we feel that it has merit but does not fully meet PLOS ONE’s publication criteria as it currently stands. Therefore, we invite you to submit a revised version of the manuscript that addresses the points raised during the review process.

We look forward to receiving your revised manuscript.

Kind regards,

Kandasamy Ulaganathan

Academic Editor

PLOS ONE

Journal Requirements:

2. Please expand the acronym “ANR” (as indicated in your financial disclosure) so that it states the name of your funders in full.

"This work was publicly funded through ANR (the French National Research Agency) under the project EVCOPAR (ANR-20-CE35-0004-01)."

"..This work was publicly funded through ANR (the French National Research Agency) under the project EVCOPAR (ANR-20-CE35-0004-01)."

Please note that funding information should not appear in the Acknowledgments section or other areas of your manuscript. We will only publish funding information present in the Funding Statement section of the online submission form. Please remove any funding-related text from the manuscript.

8. Please upload a new copy of Figure 2 as the detail is not clear. Please follow the link for more information: 

https://blogs.plos.org/plos/2019/06/looking-good-tips-for-creating-your-plos-figures-graphics/

https://blogs.plos.org/plos/2019/06/looking-good-tips-for-creating-your-plos-figures-graphics/

Reviewers' comments:

Reviewer's Responses to Questions

**Comments to the Author**

1. Is the manuscript technically sound, and do the data support the conclusions?

Reviewer #1: Yes

Reviewer #2: Partly

2. Has the statistical analysis been performed appropriately and rigorously? 

Reviewer #1: Yes

Reviewer #2: I Don't Know

3. Have the authors made all data underlying the findings in their manuscript fully available?

Reviewer #1: Yes

Reviewer #2: No

4. Is the manuscript presented in an intelligible fashion and written in standard English?

Reviewer #1: Yes

Reviewer #2: Yes

5. Review Comments to the Author

Reviewer #1: A pre-print of this manuscript entitled “Deciphering mixed infections by plant RNA virus and reconstructing complete genomes simultaneously present within-host” by these authors available on the https://www.biorxiv.org/content/10.1101/2024.02.29.582683v1.

doi: https://doi.org/10.1101/2024.02.29.582683

The manuscript well structured, and the approaches employed for nucleic acid extraction, library preparation, genome detection, bioinformatics and data analysis were deemed suitable.

However, the manuscript requires thorough editing, with specific focus on English grammar, spelling, and sentence structure.

For example, the caption of Table S2 should be revised.

In the version that I have, the resolution and quality of all graphs and figure are unsatisfactory. The authors should work on making them better.

Reviewer #2: Although the idea and the experimental design for this manuscript are interesting, the manuscript in the present form presents important lack of information and some methodological issues.

In the introduction section, the authors never comment transmission mechanisms of this virus that may favour the spreading of mixed infections.

In the methods section, methodology used for inoculations is not even mentioned in the text. It is not indicated how many samples were inoculated and what checks have been done to make possible for authors claiming specific strain prevalence in each mixed infection under analysis. It is not clear if they performed any biological and/or technical replicate either for sequencing or for inoculations. At L153-154, they speak about sample conservation, commenting a specific batch in comparison to others (?), without explaining why one was better than the other(s).

In table 1, they claim the use of the two nanopore flowcells (R9 and R10), which have different chemistry leading to different error rates, but this has never been mentioned neither commented in the text. Please, provide an explanation in the text on the reason why you changed chemistry.

The search of recombinants has been done with RDP, but no information has been provided concerning the reference sequence dataset nor the parameters used. It is important to run the RDP analysis using all the full length sequences available, since the results are always biased by the reference dataset.

As a consequence of the lack of clarity in the methods section the results appear confused as well, therefore needing a reorganization in line with the information provided in the previous sections.

The discussion needs to be updated in agreement with the modifications of methods and results sections. In addition, L342-343, the authors claim that the methodology they describe "does not capture the viral clouds...", I think that this is not true. Nanopore sequencing is a single molecule sequencing technology, therefore if the authors were able to sequence even just a handful of complete(or almost) genomes, if the error rate is not to high, this could be considered as the viral cloud contained in that specific sample. This check could be particular useful in the case of libraries obtained with R10 chemistry.

L352-353: the claim that nanopore is cost-effective is not totally true. The starter kit could be cheaper that buying an Illumina apparatus, but in terms of library cost, together with the "cheap" nanopore kits, the users have to buy quite a lot of other reagents that, in the end, does not make the sequencing so cheaper compared to an outsourced illumina sequencing. Not to mention the possibility of making mistakes when preparing libraries compared to Illumina standardized systems.

Finally, the authors did not share any information about the sequences they obtained, I think that if accession numbers are not available they could have at least confidentially shared the sequences identified.

Table 1. Section "Sequencing information", no data provided for the QC of reads. For the "RVhaplo results", please provide unit measures. In the column "distance to reference" please specify the unit measure for this parameter and what is the meaning of the strain within parentheses.

Figure 1. Panel of VNPrimers, homogenize to decimal numeration.

Figure 2. figure with illegible text

6. PLOS authors have the option to publish the peer review history of their article (what does this mean?). If published, this will include your full peer review and any attached files.

Reviewer #1: No

Reviewer #2: No

---

## [Author Response · Author response to Decision Letter 0]

26 Jun 2024

We thank the 2 reviewers for their contribution to improve the manuscript

The response to all comments raised by the reviewers appear in the specific doc file

---

## [Decision Letter · Decision Letter 1]

18 Jul 2024

PONE-D-24-16411R1Deciphering mixed infections by plant RNA virus and reconstructing complete genomes simultaneously present within-hostPLOS ONE

Dear Dr. Tollenaere,

Thank you for submitting your manuscript to PLOS ONE. After careful consideration, we feel that it has merit but does not fully meet PLOS ONE’s publication criteria as it currently stands. Therefore, we invite you to submit a revised version of the manuscript that addresses the points raised during the review process.

We look forward to receiving your revised manuscript.

Kind regards,

Kandasamy Ulaganathan

Academic Editor

PLOS ONE

Journal Requirements:

Reviewers' comments:

Reviewer's Responses to Questions

**Comments to the Author**

1. If the authors have adequately addressed your comments raised in a previous round of review and you feel that this manuscript is now acceptable for publication, you may indicate that here to bypass the “Comments to the Author” section, enter your conflict of interest statement in the “Confidential to Editor” section, and submit your "Accept" recommendation.

Reviewer #1: All comments have been addressed

Reviewer #2: All comments have been addressed

2. Is the manuscript technically sound, and do the data support the conclusions?

Reviewer #1: Yes

Reviewer #2: Yes

3. Has the statistical analysis been performed appropriately and rigorously? 

Reviewer #1: Yes

Reviewer #2: I Don't Know

4. Have the authors made all data underlying the findings in their manuscript fully available?

Reviewer #1: Yes

Reviewer #2: Yes

5. Is the manuscript presented in an intelligible fashion and written in standard English?

Reviewer #1: Yes

Reviewer #2: Yes

6. Review Comments to the Author

Reviewer #1: The authors adequately addressed the reviewers' comments and improved the manuscript accordingly.

However, in numerous cases, the lines referenced by the authors in their response to the referee were incorrect, causing difficulties for the reviewing.

I think this manuscript may be published in “PLOS ONE” Journal. However, some comments should be addressed before acceptance:

Page 4, Lines 75-78: "The word 'RYMV' was repeatedly used at the beginning of each sentence. Please revise the sentences to avoid starting each one with 'RYMV.'" In addition, remove the comma after "Africa"; replace "in" with "into" for "organized into five overlapping open reading frames"; change "was found to harbor a particularly high RYMV genetic diversity" to "was found to harbor particularly high RYMV genetic diversity"; add "several" before "years".

The paragraph should be revised as follows: Rice yellow mottle virus (RYMV, genus Sobemovirus) causes an endemic rice disease in Africa that poses a threat to the sustainable development of rice production on the continent [30]. The virus is transmitted mechanically, with agricultural practices playing an important role, as well as through biotic means, including various insect vectors [31]. RYMV is a (+) single-strand RNA virus, with a genome of approximately 4450 nt, lacking a 3’ polyA tail and organized into five overlapping open reading frames [30]. A recent study identified a rice yellow mottle disease hotspot in Burkina Faso [32, 33]. This irrigated site was found to harbor particularly high RYMV genetic diversity, with at least four distinct genetic groups coexisting over several years [33].

Page 6, Lines 119-125: This paragraph lacks clarity and needs to be rewritten for better understanding. I recommend it be revised as follows:

Each of the three methods (VN primers, RYMV-specific primers and direct cDNA) was performed twice, using RNA extracted from rice plant infected with the BF710 isolate. This infected material came from experimental infections carried out mechanically under controlled conditions [34], using symptomatic rice leaves collected two to five weeks post-inoculation in greenhouses. This approach for viral replication was employed for all samples referred to as “isolates.” The initial inoculum consisted of rice leaves collected from fields in southwestern Burkina Faso. During field surveys, farmers were individually asked for permission to sample leaves from their fields. The entire project adhered to the Nagoya protocol guidelines on access and benefit-sharing.

Please remove the vertical lines from the “Table 1”.

-In Table S2: A real recombination event can be detected with high confidence using multiple different methods and with a low associated p-value for each of the methods.

Generally, a recombination event was credibly accepted when detected by three or more methods with p-values < 1.0 × 10–6.

I think the evidence for the second recombination event is not strong or conclusive enough, necessitating further research to confirm its occurrence or understand its significance.

Did the authors consider using other software to verify the recombination event? For example, construct tree using SplitTree software including recombinants.

Reviewer #2: The corrections made improved the clarity of the manuscript. Please find below some minor suggestions.

L52-53: VANA (Virion-associated nucleic acid)-based

L98: full-length

L168-169: to one known RYMY isolate singly sequenced (runs 6 and 7). Please carefully recheck table caption for English

Table 1, first column header "Run nb", please recheck and correct.

Figure 2 caption: please recheck English and text, particularly the figure title

L429-431: text not clear, please reword

S2 table: n° Recombination event, change to "recombination event". For columns "Beginning breakpoint" and "ending breakpoint", please indicate the meaning of numbers within squared parenthesis

7. PLOS authors have the option to publish the peer review history of their article (what does this mean?). If published, this will include your full peer review and any attached files.

Reviewer #1: No

Reviewer #2: No

---

## [Author Response · Author response to Decision Letter 1]

12 Sep 2024

Dear reviewers,

We thank you for the number of suggestions leading to improve our research article. 

This led to a significant improved manuscript, while the conclusions did not change.

All the comments and corrections proposed by the two reviewers were addressed, as described in the document ("response to reviewers").

We also added the genbank accession numbers to the S1 Table.

---

## [Decision Letter · Decision Letter 2]

23 Sep 2024

Deciphering mixed infections by plant RNA virus and reconstructing complete genomes simultaneously present within-host

PONE-D-24-16411R2

Dear Dr. Tollenaere,

We’re pleased to inform you that your manuscript has been judged scientifically suitable for publication and will be formally accepted for publication once it meets all outstanding technical requirements.

Kind regards,

Kandasamy Ulaganathan

Academic Editor

PLOS ONE

Additional Editor Comments (optional):

Reviewers' comments:

Reviewer's Responses to Questions

**Comments to the Author**

1. If the authors have adequately addressed your comments raised in a previous round of review and you feel that this manuscript is now acceptable for publication, you may indicate that here to bypass the “Comments to the Author” section, enter your conflict of interest statement in the “Confidential to Editor” section, and submit your "Accept" recommendation.

Reviewer #1: All comments have been addressed

2. Is the manuscript technically sound, and do the data support the conclusions?

Reviewer #1: Yes

3. Has the statistical analysis been performed appropriately and rigorously? 

Reviewer #1: Yes

4. Have the authors made all data underlying the findings in their manuscript fully available?

Reviewer #1: Yes

5. Is the manuscript presented in an intelligible fashion and written in standard English?

Reviewer #1: Yes

6. Review Comments to the Author

Reviewer #1: The reviewers’ remarks have been taken into consideration, and the revised manuscript may be accepted for publication in “PLOS ONE” Journal. However, a few minor typographical errors have been identified. The authors should revise the following terms before acceptance:

Line 74: The genus name (Sobemovirus) should be italicized. Please correct it.

Line 220: Insert a space between the word and number in "Fig1" or change it to "Figure 1".

Line 367: Change “structuration” to “structuring”.

7. PLOS authors have the option to publish the peer review history of their article (what does this mean?). If published, this will include your full peer review and any attached files.

Reviewer #1: No

---

## [Editor Report · Acceptance letter]

1 Oct 2024

PONE-D-24-16411R2 

PLOS ONE

Dear Dr. Tollenaere, 

I'm pleased to inform you that your manuscript has been deemed suitable for publication in PLOS ONE. Congratulations! Your manuscript is now being handed over to our production team.

Kind regards, 

on behalf of

Dr. Kandasamy Ulaganathan 

Academic Editor

PLOS ONE